# Rheological Behavior and Microstructure Characteristics of SCC Incorporating Metakaolin and Silica Fume

**DOI:** 10.3390/ma11122576

**Published:** 2018-12-18

**Authors:** Gang Ling, Zhonghe Shui, Tao Sun, Xu Gao, Yunyao Wang, Yu Sun, Guiming Wang, Zhiwei Li

**Affiliations:** 1School of Materials Science and Engineering, Wuhan University of Technology, Wuhan 430070, China; linggang0609@163.com (G.L.); yywang@whut.edu.cn (Y.W.); yuyuyuxiaomogu@aliyun.com (Y.S.); chunchun2222017@outlook.com (Z.L.); 2State Key Laboratory of Silicate Materials for Architectures, Wuhan University of Technology, Wuhan 430070, China; zhshui@whut.edu.cn (Z.S.); guimingw@hotmail.com (G.W.); 3School of Civil Engineering and Architecture, Wuhan University of Technology, Wuhan 430070, China; 17702768017@163.com

**Keywords:** self-compacting concrete (SCC), rheology, workability, pozzolanic reaction, microstructure

## Abstract

This study explores the effects of metakaolin (MK) and silica fume (SF) on rheological behaviors and microstructure of self-compacting concrete (SCC). The rheology, slump flow, V-funnel, segregation degree (SA), and compressive strength of SCC are investigated. Microstructure characteristics, including hydration product and pore structure, are also studied. The results show that adding MK and SF instead of 4%, 6% and 8% fly ash (FA) reduces flowability of SCC; this is due to the fact that the specific surface area of MK and SF is larger than FA, and the total water demand increases as a result. However, the flowability increases when replacement ratio is 2%, as the small MK and SF particles will fill in the interstitial space of mixture and more free water is released. The fluidity, slump flow, and SA decrease linearly with the increase of yield stress. The total amount of SF and MK should be no more than 6% to meet the requirement of self-compacting. Adding MK or SF to SCC results in more hydration products, less Ca(OH)_2_ and refinement of pore structure, leading to obvious strength and durability improvements. When the total dosage of MK and SF admixture is 6%, these beneficial effects on workability, mechanical performance, and microstructure are more significant when SF and MK are applied together.

## 1. Introduction

Self-compacting concrete (SCC) is usually characterized as a high-performance concrete that can pass through the gaps between steel bars and fill the formwork completely, only relying on its own gravity during pouring process [1,2,3]. It is distinguished by excellent workability; thus, no vibration is required during casting, which can significantly reduce the cost, simplify the protocol, shorten the construction time, and guarantee the homogeneity of concrete, especially when applied to the complex cross-section structures [4,5]. Owing to those advantages, increasing attention has been paid to SCC since it was first prepared in 1988 [6]. The workability of self-compacting concrete is assessed by both fluidity and homogeneity. However, a high fluidity is usually accompanied with poor homogeneity, and SCC is sensitive to its manufacturing parameters. Therefore, balancing the fluidity and homogeneity is crucial in the design of SCC [7,8].

The rheological properties of paste have significant influence on the workability of SCC [9,10,11,12,13]. Many studies have been carried out to study the rheological properties of SCC on levels of concrete, mortar, and paste. It is generally accepted that the rheology of SCC conforms to the Bingham model, and governed by two fundamental parameters: the yield stress and viscosity [14,15,16]. Yield stress is the minimum force to initiate the flow of SCC, a series of experiments on the relationship between slump and yield stress were conducted, and the results showed that there was an inverse proportional function between yield stress and fluidity [17,18]. A previous study on the influence of mortar rheology on homogeneity showed that yield stress is the key factor to prevent the segregation of aggregate, and the yield stress should be higher than the lowest critical value to ensure the suspension stability [19]. It was also reported that the flow rate of paste is determined by viscosity. The workability of SCC can be partly predicted by analyzing the rheological properties of paste and mortar [20]. For instance, the Krieger–Dougherty (K–D) formula is proven to be able to simulate the correlation of viscosity and yield stress between the mortar and paste [21], the relationship between rheological properties of SCC and paste was established based on the K–D formula and mortar layer model, and a new method for design of SCC based on net paste rheology is provided [22]. Therefore, studying the rheology is of great significance during the design of SCC.

On the other hand, supplementary cementitious material (SCM) has become an important component in the design of modern SCC [23]. Fly ash (FA) is one of the most widely used SCMs in SCC for its beneficial effect on workability. It is found that SCCs that contain 30%–60% FA have good mechanical properties and durability [24]. Moreover, the addition of fly ash to recycled aggregates concrete will improve its workability, compressive/tensile strengths, and resistance to chloride [25,26]. Nevertheless, high-volume FA content also increases the risk of segregation and reduces early strength [27]. Silica fume (SF) is commonly used in concrete to improve the stability and mechanical properties [28,29]. Adding 20% silica fume to SCC was reported to increase the compressive strength by 27% at 28 days [30]. Meanwhile, metakaolin (MK) is a binding material with similar particle size and pozzolanic activity to SF, and has been widely used in concrete [31,32,33,34,35]. Performances, such as mechanical properties and durability, can be remarkably improved after MK is included [36,37]. SF and MK addition in SCC will increase the stability on the one hand, but negatively affect the fluidity on the other hand; the water demand of concrete is significantly increased with SF and MK addition. The poor fluidity will inhibit the discharging of bubbles and damage the filling performance during construction, which is harmful to the general performance of SCC [38,39]. Based on the problems above, it indicates that the use of appropriate compounded MK and SF is a novel approach to develop ultra-high-performance concrete (UHPC) with advanced properties, and the optimum dosage is 3% MK + 5% SF. The reason is that the utilized MK is more active than SF, but an excess amount of MK can increase the viscosity and shrinkage, leading to trapped bubbles and microcracks [40]. Hence, it is logical to study the feasibility of producing SCC by composite use of MK and SF.

The preparation of high-performance SCC usually requires the addition of SF and MK admixtures, but the negative influence on the workability is also a deterrent. Hence, it is of great significance to propose a design criterion of high-performance SCC with consideration of SF and MK. In this study, the effect of SF and MK on the rheology and workability of high-performance SCC is investigated, and importance of the rheology on the workability of SCC is discussed. Then, the enhanced mechanical properties and microstructure of high-performance SCC containing SF and MK admixtures are evaluated.

## 2. Materials and Methods

### 2.1. Materials

In this study, P·II 42.5 cement is used as binding material; fly ash (FA), silica fume (SF), and metakaolin (MK) are used as SCMs, their particle size distributions and chemical constituents are measured by laser particle size analyzer and X-ray fluorescence spectrometer, respectively, as shown in Figure 1 and Table 1, respectively. Natural river sand with fineness modulus of 2.9 is used as fine aggregate. Continuous graded gravels with a size of 5–20 mm are used as coarse aggregate. A polycarboxylate superplasticizer (SP) with a solid content of 20% is employed to adjust the flowability of SCC.

### 2.2. Mix Design

The mix composition of paste and concrete mixtures are shown in Table 2 and Table 3, respectively. Cement accounts for 55% of the total cementitious material and the rest are SCMs. FA is partially replaced by SF and MK to improve the performance of SCC. The water/binder (w/b) ratio is fixed at 0.33 for all mixes. Paste specimens are prepared according to ASTM C 305-2006 and concrete specimens are prepared in accordance with ASTM C192 [41]. Cubic specimens of 100 × 100 × 100 mm^3^ are prepared, and demoulded after 24 h, then they are cured under standard conditions (25 ± 2 °C and relative humidity of 98%) until testing.

### 2.3. Testing Methods

#### 2.3.1. Rheology Behavior

The rheological property of paste is measured by a rotor viscometer. The testing instrument is R/S-SST soft-solid rheometer produced by Brookfield Company (Toronto, Canada). The range of shear stress is 6–200 Pa and the shear rate is 0–1000 rpm. The data obtained by the rheometer can be transmitted to the master computer in real time and analyzed by Rheo V2.8 software. The rheological test procedure of rheometer is shown in Figure 2.

#### 2.3.2. Workability

In order to evaluate the workability of SCC, slump flow test and V-funnel test were carried out. Segregation degree of coarse aggregate (SA) was also measured to evaluate the homogeneity of SCC. The tests process was referred to BS EN 12350:2010 [42]. SA was defined as the difference of coarse aggregate weight in different layers: Firstly, the three-layer segregation barrel containing SCC is placed on the jumping table, then the jumping table is started and vibrated 25 times. The concretes in each layer are put into the 5 mm sieves. After washing and drying, the quality of aggregates in different layers is weighed. The SA can be calculated as shown in Equation (1).
(1)$$\mathrm{SA}=\frac{m_{1}-m_{2}}{m_{0}},$$
where *m*_0_ is the average weight of dried aggregate in concrete, and *m*_1_ and *m*_2_ are the weight of dried aggregates in the bottom and top layer of the segregation barrel, respectively.

#### 2.3.3. Compressive Strength

A 3000 kN capacity machine was employed to measure the compressive strength of cube specimens according to ASTM C39 [43]. The test was conducted at the ages of 3 days and 28 days. Three specimens for each batch were measured and the average value was employed to evaluate the compressive strength.

#### 2.3.4. Hydration Products

The samples of paste were crushed into small grains after curing for 28 days, and soaked in alcohol to stop hydration. Then, the samples were dried at 60 °C for 24 h in an oven. Afterwards, the samples were grounded into powder and examined by XRD and thermal analysis.

XRD analysis is carried out by D/max-RB type X-ray diffractometer (Japanese RIGAKU, Tokyo, Japan) with scanning step of 0.02, the speed of 10 °/min, the current of 30 mA, and the voltage of 35 kV. Thermal analysis is carried out by Simultaneous Thermal Analyzer (Netzsch STA449F3, Shanghai, China) under nitrogen atmosphere with gas flow of 30 mL/min, heating rate of 10 °C/min, and temperature range of 0~1000 °C.

#### 2.3.5. Pore Size Distribution

The small grains with several micrometers in diameter, crushed from 28-day concrete samples, were dried in an oven at 60 °C for 24 h before pore structure examination. The Micromeritics AutoPore-9500 (Micromeritice, Norcross, GA, USA) was used for mercury intrusion porosimetry test (MIP) test, with the generating pressures of mercury intrusion porosimeter ranging from 3 Pa to 379 MPa.

## 3. Results and Discussions

### 3.1. Rheology Behavior

The effect of MK and SF on the fluidity of pastes is shown in Table 4, and it can be seen that fluidity of the paste decreases gradually with the increase of metakaolin and silica fume content. This is due to that the specific surface area of MK and SF is obviously larger than that of fly ash (as shown in Table 1); and the total water demand increases as a result. The free water content in paste is reduced, and the fluidity declined. Compared with P0, the fluidity of paste decreases by 30.9% for 8% MK addition, and 40% for 8% SF addition, respectively, which indicates that SF has greater influence than MK on fluidity. This is because the particle size of SF is smaller as shown in Figure 1, therefore, more water is adsorbed by the powder itself.

Figure 3 shows the rheological curve of PMK6. It can be seen that the rheological curve is mainly divided into two parts, corresponding to the second and third parts of the rheological program (as shown in Figure 2). The curve of the up part is irregular, and this phenomenon can be attributed to the reason that the paste is solid-like, and contains the local “weak” regions at the beginning of shearing [44], and pre-shearing will liquefy the paste, but not completely. The down part shows a regular linear relationship, indicating that the paste has been completely liquefied, which is mainly due to the thixotropy of the paste. The analysis of rheological test is mainly focused on the down part, as shown in Figure 3. The Origin Pro 2015 is employed to perform the procedure of linear fitting. The fitting curve of PMK6 is y = 1.384x + 13.913, *R*^2^ is 0.99939. It shows that the mathematic relationships between shear stress (τ) with shear rate (γ) of pastes can be determined from being fitted by Bingham model (τ = τ_0_ + η·γ), where slope of the curve, η, represents the viscosity and the vertical intercept; τ_0_, represents the yield stress of the paste [45].

The viscosity and yield stress of different pastes are shown in Table 4 and Figure 4. It can be seen from the Table 4 and Figure 4 that the plastic viscosity and yield stress of the paste increase gradually with increasing contents of MK and SF, which can explain the decrease of paste fluidity. When the content of MK and SF is 2%, the viscosity and yield stress of PMK2 is 38.2% and 86.3% lower than P0, respectively. The reduction of viscosity and yield stress is 8.7% and 70.6% for PSF2, respectively. This is due to the small particle size of MK and SF, which can be filled in the interstitial space between the particles of paste, thus, more free water is released. Comparing PMK8 with P0, it can be found that the viscosity of the paste is increased by 38.4% and the yield stress is increased by 5.5 times when 8% MK is added. This indicates that MK addition can be more significantly affected by the yield stress rather than viscosity. The same observation can be found when SF is added. The phenomenon could be attributed to the following reasons: on the one hand, the particle size of MK and SF are smaller than FA, and the addition of MK and SF will reduce the content of free water in the mixture; on the other hand, MK and SF are more active and will promote the hydration process, and the flocculated hydration products will wrap in the surface of the particles and prevent the relative movement of the particles, resulting in a significant increase of yield stress [46]. When the content of MK and SF is 8%, the yield stress and viscosity of PSF8 are 57.2% and 25.6% higher than that of PMK8, respectively. This indicates that SF has a larger effect on rheological properties than MK, and it is in keeping with the result of fluidity test. When the total content of MK and SF is 6%, the viscosity and yield stress of PMK2SF4 and PMK4SF2 are slightly lower than that of PMK6 and PSF6, indicating that the negative effect of MK or SF on the fluidity can be minimized by the binary use of MK and SF. This is because SF is finer than MK, when they are used in combination, and the accumulation of the powder becomes more compact and more free water is released, resulting in increased fluidity of paste.

Overall, it can be found that there is an obvious linear relationship between fluidity and yield stress, as shown in the Figure 5, and the fitting curve is y = 272 − 4.02x, and the square difference, *R*^2^, is 0.913. This demonstrates that the fluidity of paste is mainly affected by yield stress, which is consist with [17]. Hence, the yield stress of paste needs to be controlled within the appropriate range to obtain excellent fluidity.

### 3.2. Workability

The slump flow, V-flow, and SA values of SCC are shown in Figure 6. It can be seen that the slump flow and SA of SCC decrease with the increase of MK and SF content (Shown as the red arrows in Figure 6). The slump flow and SA of C0 (without MK and SF) are 650 mm and 23.4%, respectively, while the slump flow of SCC with 2% MK (CMK2) and 2% SF (CSF2) is 720 mm and 700 mm, respectively. Similarly, the SA of SCC with 2% MK (CMK2) and 2% SF (CSF2) is 25.1% and 24.1%, respectively. Hence, both MK and SF will increase the fluidity and segregation of SCC when their content is 2%. The slump flow and SA of SCC that contain SF are lower than MK, showing that SF has greater impact on the workability of SCC than MK, which follows a similar tendency as the rheological results, and it is indicated that the rheology of paste is very important to the performance of SCC. When the total dosage of SF and MK admixture is 6%, the slump flow of CMK6 (6% MK) and CSF6 (6% SF) is 630 mm and 620 mm, separately. While the slump flow of CMK2SF4 (2% MK + 4% SF) and CMK4SF2 (4% MK + 2% SF) is 635 mm and 680 mm, separately. It can be seen that the compound use of MK and SF can obtain higher fluidity with the same dosage. This is attributed to difference of particle sizes between MK and SF, then the grain composition and accumulation of powders is optimized by the combination of MK and SF. As a consequence, more free water is released, and the fluidity increased. V-flow is the result of the combined effect of fluidity and stability. Low fluidity will lead to slow flow rate, and the V-flow value will increase. However, excessive fluidity leads to the segregation of aggregate, and the aggregate will concentrate at the outlet, resulting in increased V-flow value. With the incorporation of SF and MK over 6%, the slump flow and V-flow of CMK8 (8% MK) is 580 mm and 20.2 s, and it is 520 mm and 32.2 s for CSF8 (8% SF). The results show that the workability of CMK8 and CSF8 does not meet the requirements of self-compacting concrete (slump flow 600~700 mm, V-flow 5~15 s), therefore, it is suggested that the mixture of SF and MK be controlled to within 6% without increasing the SP dosage in this study.

Figure 7 shows the effect of yield stress of paste on the slump flow of SCC. The fitting curve of yield stress and slump flow is y = 711 − 6.18x, the square difference *R*^2^ is 0.948, and it shows a significant linear relationship between yield stress and slump flow, and the slump flow decreases with the increase of yield stress. This phenomenon illustrates that the yield stress of paste is the key factor to determining the slump of SCC. The reason is that when the driving force is greater than the yield stress of SCC, the flow continues; conversely, it stops [22].

Figure 8 shows the relationship between yield stress and SA of SCC. The fitting curve is y = 20.09 − 0.581x, and *R*^2^ = 0.864. A linear relationship can also be seen between yield stress and SA of SCC. The fitting curve indicates that the SA decreases with the increase of yield stress, and the segregation of aggregate will not occur (SA = 0) when the yield stress is greater than 34.6 Pa. This phenomenon is attributed to by the fact that yield stress is the precondition for the stability of aggregate in concrete, and segregation of aggregate happens only when the difference between gravity and buoyancy of the aggregate is greater than the yield stress [47]. Nevertheless, there is some fluctuation for linear relationship between the yield stress and SA of SCC, and it indicates that SA is also affected by viscosity of concrete. During the SA test, the segregation bucket containing concrete should be placed on the jumping table and vibrated 25 times, and the yield stress will be significantly reduced under these vibration conditions and, as a result, the aggregate will fail to remain suspended and sink [19]. At this point, the sedimentation rate of aggregate is affected by the viscosity based on the Stokes formula. In summary, the rheology of paste is an important factor affecting the workability of SCC, and the performance of SCC can be predicted, to a certain extent, by evaluating the rheology property of the paste [48].

### 3.3. Compressive Strength

The compressive strength of 3 and 28 days of SCC that contain different amounts of MK and SF are shown in the Figure 9. In general, it can be seen that the addition of MK and SF can increase the compressive strength of SCC. For example, comparing with C0, the addition of 6% MK (CMK6) increases the strength of SCC by 22.3% and 15.1% at 3 and 28 days, respectively, and it is 16.0% and 17.0% for CSF6. Moreover, when the dosage of MK and SF in SCC are relatively close (e.g., CMK2SF4 and CMK4SF2), the compressive strengths are higher than that of CMK6 and CSF6. The compressive strength is 64.3MPa for CMK4SF2 at 28 days, which is the highest value in all specimens. This phenomenon can be explained by the following reasons: On the one hand, the particle size of MK and SF are smaller than FA (as shown in Figure 1) and can fill the voids between powers and, moreover, the activity of MK and SF are higher than FA. Thus, more free water in the interspace is released, and the hydration reaction is further facilitated. On the other hand, MK and SF are more active, more hydration products are generated by pozzolanic reaction and fill the voids in hardened concrete. All of these effects will lead to a denser structure and a higher strength [40]. In addition, SF is finer than SF and, when MK and SF are composited, the small MK particle can be the kernel of the early pozzolanic reaction and the formation of C-S-H (calcium silicate hydrate) gels, which can further accelerate the cement hydration and fill the pores. It should be noticed that the differences of the compressive strengths for the samples with same content of MK and SF are relatively small. The enhancement effect is more increased when MK and SF are binary applied. As the addition of MK and SF is 8% (e.g., CMK8 and CSF8), the compressive strength is lower than that of CMK6 and CSF6. It is shown in Figure 6 that the fluidity of CMK8 and CSF8 is too low, and the bubbles in the concrete are difficult to release. Thus, the compactness of hardened concrete structure will be reduced. This illustrates the importance of workability on the mechanical properties of SCC. In order to clearly clarify expound the effect of MK and SF on the mechanical properties of SCC, their microstructures are explored in the sections below.

### 3.4. Hydration Products

The XRD patterns of different mixtures after curing for 28 days are presented in Figure 10. It can be seen that the main hydration products are AFt (ettringite), Ca(OH)_2_ (marked as CH), and CaCO_3_ (marked as C). The diffraction peak of AFt in PSF6 is lower than P0, which indicates that the employment of SF decreases the formation of AFt. This should be attributed to reduction in active aluminum phases and more significant absorption of aluminum by C-S-H gels in SF system. This is in accordance with a previous study [49]. It can be clearly observed that the diffraction peak of Ca(OH)_2_ in PMK6 and PSF6 is inferior to P0, that is, the employment of MK and SF will reduce the content of Ca(OH)_2_. This is because MK and SF are more active than FA, and doping MK and SF can increase the content of high activity SiO_2_ and Al_2_O_3_ in SCC. Thus, more Ca(OH)_2_ produced by cement hydration is consumed by pozzolanic reaction and better performances, such as with mechanical properties, can result (as shown in Figure 9) [50,51,52].

DTG (derivative thermogravimetry) curves of specimens after curing for 28 days are presented in Figure 11. As TG-DSC test is conducted on samples after completely dried at 40 °C, and the state that adsorbed water is completely removed, while bound water is not separated has been achieved [53]. Therefore, the weight loss caused by water under this condition is induced by bound water.

It can be clearly found from Figure 11 that the curve peak of hydration products (including C-S-H gels, Aft, and CaCO_3_) in PMK6 and PSF6 is more remarkable than P0, which means that the addition of MK and SF induces much more abundant hydration products [54,55]. This is attributed to the accelerated hydration process caused by MK and SF incorporation [56]. The addition of MK and SF reduced the mass loss peak of Ca(OH)_2_, indicating that the content of Ca(OH)_2_ in hydration products decreases, and is accordant with the XRD results (as shown in Figure 10). The mass loss peak of AFm (hydrated calcium aluminate sulfate) can be expressly discovered for the samples of PMK6 and PMK2SF4 in Figure 11. This is deducted from formation and transformation of SO_4_-AFm. According to formation of AFt, gypsum dissolves completely in the cement-based system after 2 days. The excess aluminum phases will react with AFt to generate AFm [57]. The content of aluminum phase in MK is relatively high, so the incorporation of MK will lead to more aluminum phases. More remarkable formation of AFm means additional of aluminum phases. As a result, it can be predicted that the mixtures have a better ability to adsorb and solidify chloride, which will lead to preferable durability in chloride-eroded environments (e.g., marine environment).

### 3.5. Pore Structure

The pore structure of concrete specimens after curing for 28 days is presented in Figure 12 and Table 5. In general, the employment of MK and SF optimizes the pores structures of SCC by reducing the total porosity and the volume of coarse pores [35]. For instance, the pore distribution of C0 is mainly in the range of 10–70 nm, but it is 10–50 nm for CMK6 and CSF6. In addition, incorporating 6% MK and SF can reduce the content of coarse pores (>50 nm) by 18.2% and 13.6%, respectively. Furthermore, both the average diameter and most probate pore diameter are reduced. Hence, the addition of MK and SF is beneficial for refining the pore structure of SCC. Due to the fact that MK and SF are finer and more active, the small MK and SF particles will fill the gaps in the powders, they can also promote the cement hydration and fill the pores, resulting in denser microstructure.

Compared with CSF6, samples that contain 6% MK (CMK6) have a relatively lower coarse pore volume and smaller average diameter, which means that MK has better capacity to refine pores, and a similar phenomenon has been observed in previous research [35]. However, it is worth noting that the mixture of CMK2SF4 has smaller pore volume than CMK6 and CSF6. Meanwhile, the mixture with compounded MK and SF (CMK2SF4) reduces the coarse pores by 50%. In addition, its average diameter and most probate pore diameter are15.4 nm and 26.3 nm, respectively, which are the lowest value of the all samples. This means that the pore structure is further optimized when MK and SF are used together. Consequently, applying a SF–MK mixture with proper ratios and contents can be a better method to develop high-performance SCC. These results are consistent with analysis of compressive strength tests [58,59,60].

## 4. Conclusions

High performance SCC with MK and SF addition are designed and characterized. The effect of MK and SF on the rheology, workability, compressive strength, and microstructure is investigated. The following conclusions can be drawn based on the results:

The employment of MK and SF, instead of FA, increases the yield stress and viscosity of SCC, resulting in the reduction of fluidity, slump flow, and SA of SCC, which is due to that the utilized MK and SF in this study are finer and more active than FA, so more water is adsorbed by the powder itself and more flocculating products are produced. When the dosage of MK and SF is 2%, the small particles will fill in the interstitial space of mixtures, and more free water is released, leading to an increasement of fluidity. The total amount of SF and MK should be no more than 6% to meet the requirement of self-compacting.

The fluidity, slump flow and SA decreases linearly with the increase of yield stress, based on rheology and workability results. This is owing to that yield stress is the key factor determining flow and aggregate settlement of concrete.

The addition of 6% MK and 6% SF instead of FA increases the compressive strength of SCC by 15.1% and 17.0% at 28 days, respectively. On the one hand, the particle size of MK and SF are smaller than FA and can fill the voids between powers. On the other hand, MK and SF are more active, and more hydration products are generated by pozzolanic reaction and fill the voids in the hardened concrete. All of these effects will lead to an optimized pore structure and higher strength. The compressive strength decreases when the contents exceed 8%, and this is due to the excessive yield stress and viscosity, which make it difficult to discharge bubbles and form a compact structure.

The property of SCC, including workability, mechanical performance, and microstructure, could be further improved when SF and MK are applied together. As the particle size of SF is smaller than MK, the accumulation of the powder becomes more compact by compounding MK and SF and, as a result, more free water is released and fluidity is increased. Moreover, the small MK particle can be the kernel of the early pozzolanic reaction and the formation of C-S-H gels, which can further accelerate the cement hydration and fill the pores. Hence, the mechanical performance and microstructure of SCC are improved.

## Figures and Tables

**Figure 1 materials-11-02576-f001:**
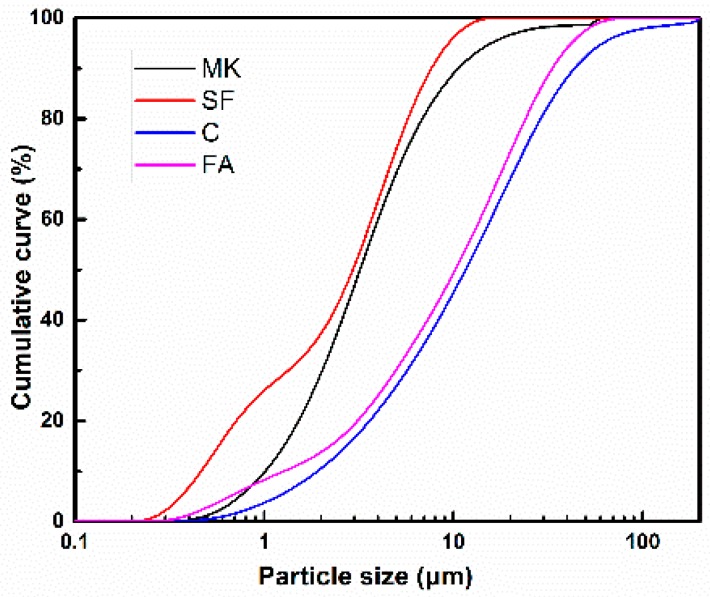
Particle size distribution curve of raw materials.

**Figure 2 materials-11-02576-f002:**
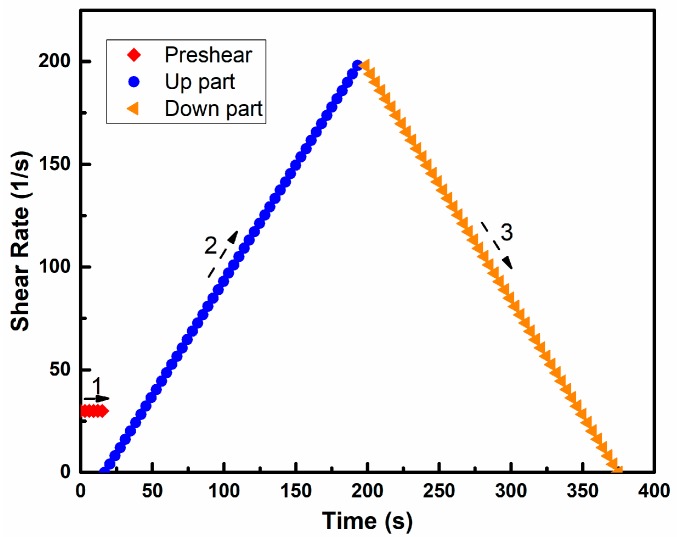
Rheological test procedure of rheometer.

**Figure 3 materials-11-02576-f003:**
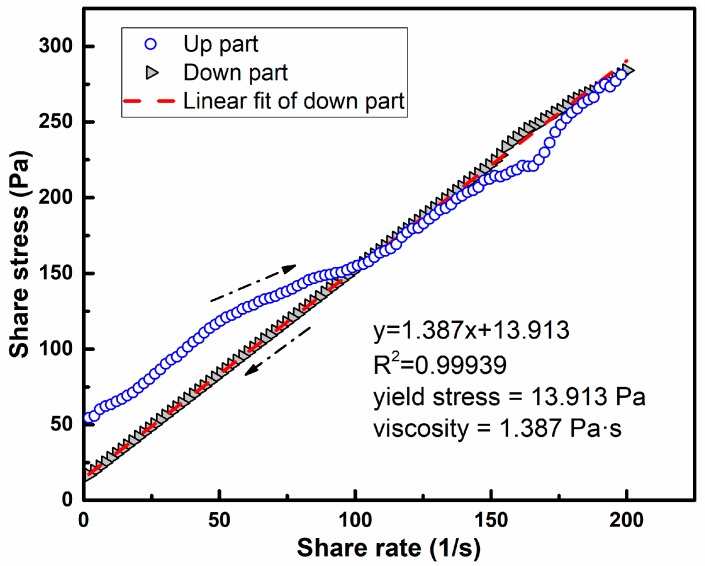
Rheological test results of PMK6.

**Figure 4 materials-11-02576-f004:**
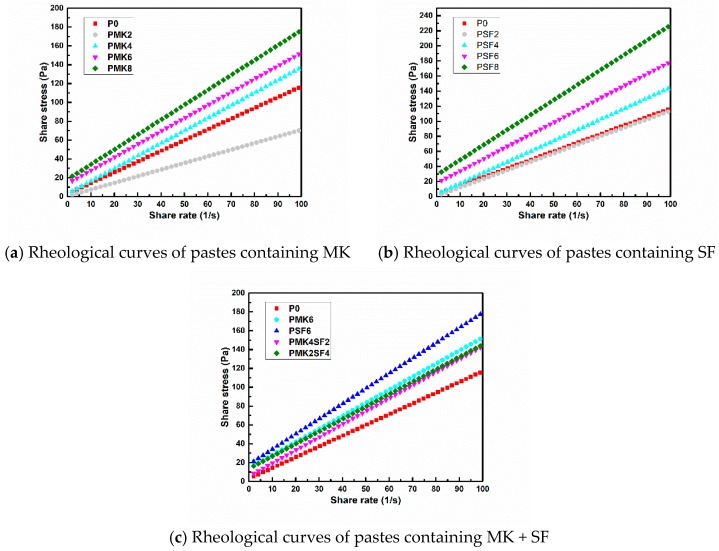
Rheological curves of pastes containing metakaolin (MK) and silica fume (SF).

**Figure 5 materials-11-02576-f005:**
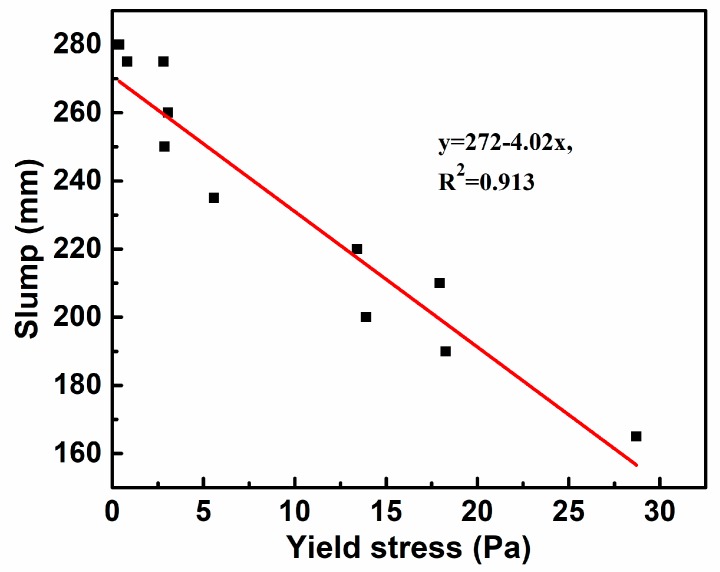
Relationship between fluidity and yield stress of paste.

**Figure 6 materials-11-02576-f006:**
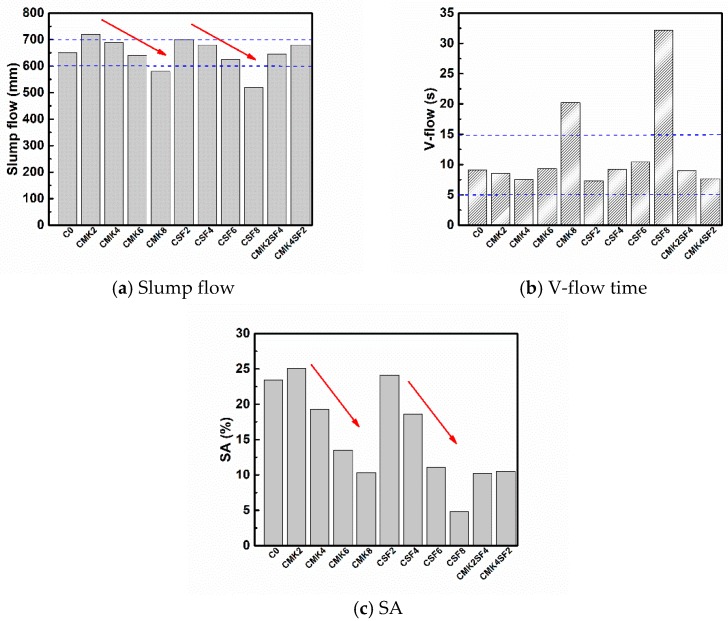
Effect of MK and SF on the workability of self-compacting concrete (SCC) (The red arrows indicate the change of SCC performance with the increase of MK and SF contents).

**Figure 7 materials-11-02576-f007:**
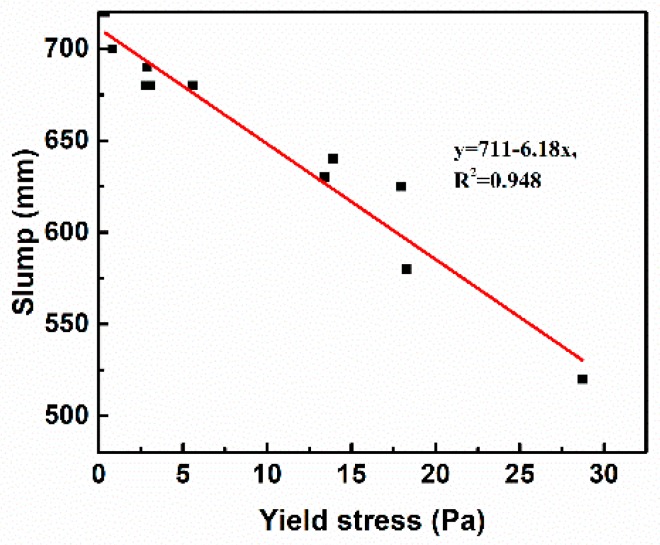
Relationship between yield stress and slump flow of SCC.

**Figure 8 materials-11-02576-f008:**
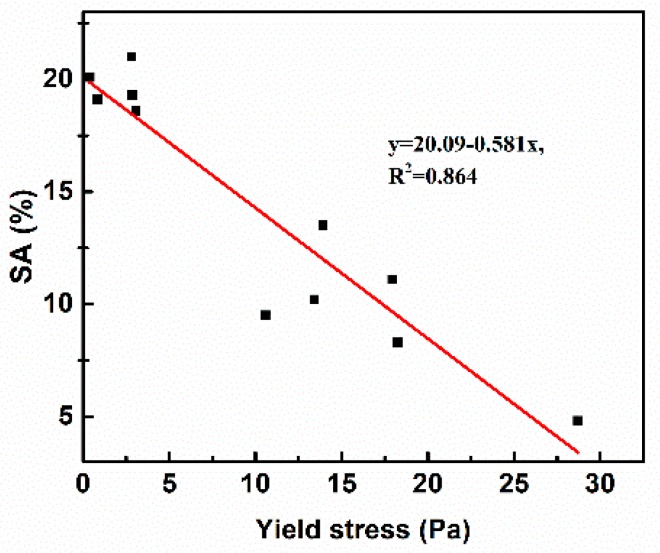
Relationship between yield stress and SA of SCC.

**Figure 9 materials-11-02576-f009:**
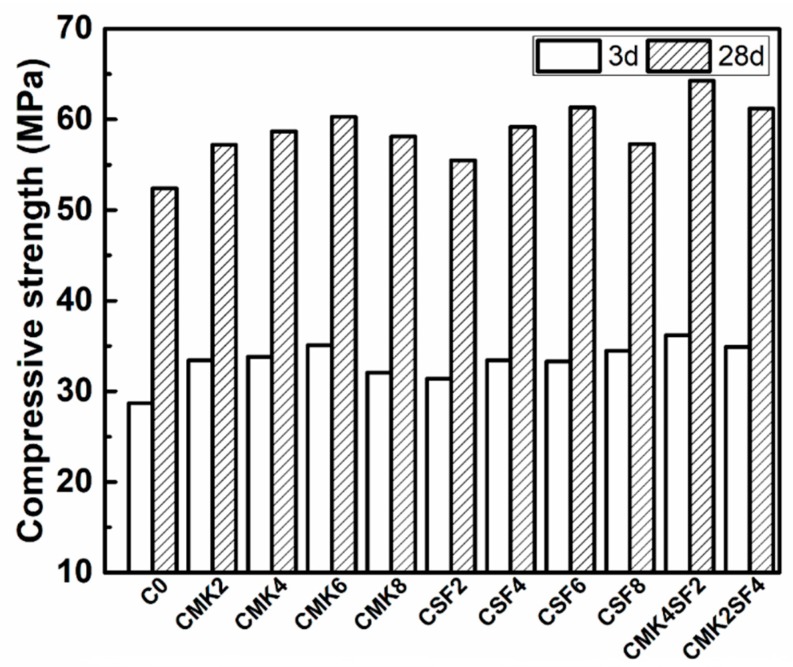
Effect of MK and SF on the compressive strength of SCC.

**Figure 10 materials-11-02576-f010:**
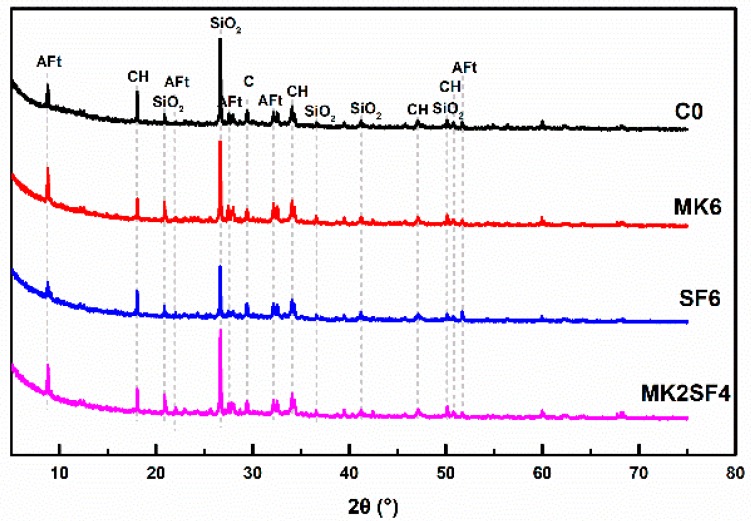
XRD patterns of specimens after curing for 28 days.

**Figure 11 materials-11-02576-f011:**
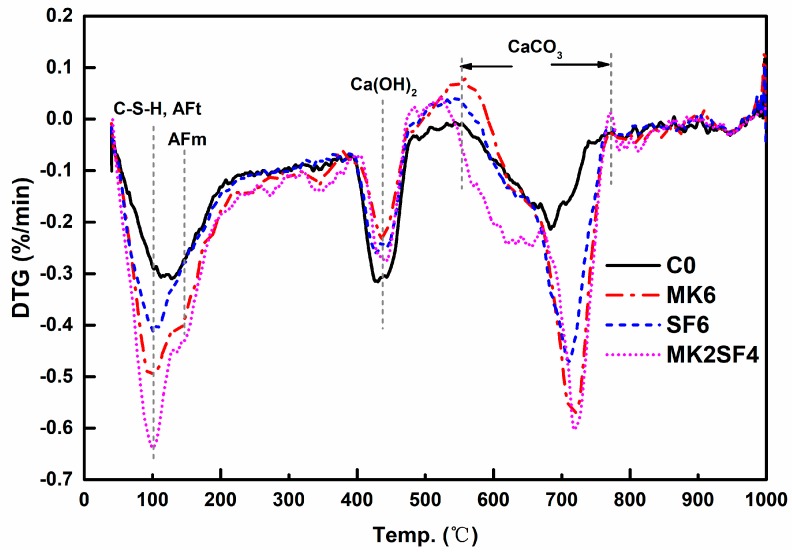
DTG (derivative thermogravimetry) curves of specimens after curing for 28 days.

**Figure 12 materials-11-02576-f012:**
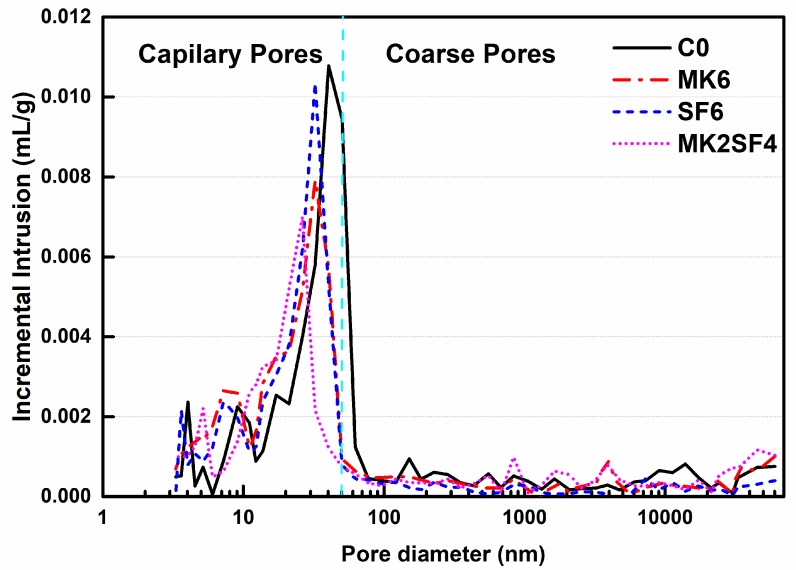
Pore structure analysis of SCC after curing for 28 days.

**Table 1 materials-11-02576-t001:** Chemical composition of powders in this study (wt %).

Compositions	SiO_2_	Al_2_O_3_	Fe_2_O_3_	CaO	MgO	K_2_O	Na_2_O	SO_3_	LOI	Specific Surface Area (m^2^/kg)
Cement	21.86	4.45	2.35	63.51	1.67	0.55	0.26	2.91	1.89	353
Fly Ash	46.43	38.02	3.11	7.51	0.23	0.89	0.34	0.68	2.78	372
Metakaolin	52.27	44.58	0.70	0.02	0.13	0.34	0.53	0.22	1.02	14,600
Silica Fume	94.65	0.15	0.25	0.33	0.49	0.85	0.16	0.66	2.21	46,100

**Table 2 materials-11-02576-t002:** Mix design of paste (kg/m^3^).

NO.	Water	Cement	Fly Ash	Metakaolin	Silica Fume	Superplasticizer
P0	552	920	753	0	0	10
PMK2	552	920	720	33	0	10
PMK4	552	920	687	66	0	10
PMK6	552	920	654	99	0	10
PMK8	552	920	621	132	0	10
PSF2	552	920	720	0	33	10
PSF4	552	920	687	0	66	10
PSF6	552	920	654	0	99	10
PSF8	552	920	621	0	132	10
PMK2SF4	552	920	654	66	33	10
PMK4SF2	552	920	654	33	66	10

**Table 3 materials-11-02576-t003:** Mix design of self-compacting concrete (SCC) (kg/m^3^).

NO.	Water	Cement	Fly Ash	Metakaolin	Silica Fume	Sand	Gravel	Superplasticizer
C0	165	275	255	0	0	859	762	5
CMK2	165	275	215	10	0	859	762	5
CMK4	165	275	205	20	0	859	762	5
CMK6	165	275	195	30	0	859	762	5
CMK8	165	275	185	40	0	859	762	5
CSF2	165	275	215	0	10	859	762	5
CSF4	165	275	205	0	20	859	762	5
CSF6	165	275	195	0	30	859	762	5
CSF8	165	275	185	0	40	859	762	5
CMK2SF4	165	275	195	10	20	859	762	5
CMK4SF2	165	275	195	20	10	859	762	5

**Table 4 materials-11-02576-t004:** Effect of Silica Fume (SF) and Metakaolin (MK) on the rheology of paste.

No.	Metakaolin (%)	Silica Fume (%)	Slump (mm)	Bingham Equation	Viscosity (Pa·s)	Yield Stress (Pa)
P0	0	0	275	τ = 1.140γ + 2.804	1.140	2.804
PMK2	2	0	280	τ = 0.704γ + 0.385	0.704	0.385
PMK4	4	0	250	τ = 1.341γ + 2.858	1.341	2.858
PMK6	6	0	200	τ = 1.387γ + 13.913	1.387	13.913
PMK8	8	0	190	τ = 1.582γ + 18.266	1.582	18.266
PSF2	0	2	275	τ = 1.130γ + 0.823	1.130	0.823
PSF4	0	4	260	τ = 1.416γ + 3.061	1.416	3.061
PSF6	0	6	210	τ = 1.611γ + 17.939	1.611	17.939
PSF8	0	8	165	τ = 1.988γ + 28.711	1.988	28.711
PMK4SF2	4	2	220	τ = 1.379γ + 5.568	1.379	5.568
PMK2SF4	2	4	225	τ = 1.318γ + 13.411	1.318	13.411

**Table 5 materials-11-02576-t005:** Pore structure of SCC after curing for 28 days.

Sample Code	C0	MK6	SF6	MK2SF4
Coarse pores (mL/g)	0.022	0.018	0.019	0.010
Capillary pores (mL/g)	0.044	0.038	0.038	0.044
Total porosity (%)	11.66	10.49	10.58	9.92
Average diameter (nm)	24.4	18.0	20.0	15.4
Most probate pore diameter (nm)	40.3	32.4	32.4	26.3

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
