# Peer review of "Rheological Behavior and Microstructure Characteristics of SCC Incorporating Metakaolin and Silica Fume"

_materials, 2018, doi:10.3390/ma11122576_

Round 1

Reviewer 1 Report

The manuscript addresses a subject of current relevance in the field of building materials technology. Therefore, this reviewer recommends it be considered for publication.

Few (minor) comments are reported throughout the annotated manuscript attached at this review report.

Author Response

Response to Reviewer 1 Comments

Piont 1: Line 59. The potential of using FA in combination with recycled aggregates should also be mentioned:Faella et al (2016), Cement and Concrete Composites, 71, 85-96. Lima et al (2013), Construction and Building Materials 47, 547-559.

Response 1: Thank you for your valuable comment. The review is right. In the revised manuscript, the potential of using FA in combination with recycled aggregates has been mentioned on the basis of reading in-depth of the two references. The addition of fly ash leading to an improvement of workability, compressive and tensile strengths, resistance to chloride penetration in concrete made with recycled aggregates.

Piont 2: Line 90. Cement accounts for 55% is tis percentage simply defined in weight? Does any "k" factor is considered for FA and the or other SCM? Please, clarify.

Response 2: Thank you for your useful comment. The review is right and deserves more consideration. Cement accounts for 55% by mass and it is marked in the manuscript. "k" factor is a matter that should be considered for FA and the other SCMs in the mix design of concrete. However, many scholars have ignored "k" factor when they study the effect of fly ash and other SCMs on the performance of concrete, and the replacement rates of SCMs are tis percentage simply defined in weight. The incorporation of FA and other SCMs in this paper refers to the experimental design method of the following references: Siddique (2004), Cement & Concrete Research, 34(3), 487-493. Yasmin et al (2018), Construction and Building Materials, 172 (30), 728-734.

Piont 3: Conclusions. This section should be completely rewritten. After an introductory sentence, the main findings have to be listed in concise items. Please, do not refer to mere empirical evidence, but try to point out the mechanical reasons behind them.

Response 3: Thank you for your useful comments and valuable suggestions on our manuscript. In the revised manuscript, the conclusions have been completely rewritten. The main findings have been listed in concise items, and the mechanical reasons have been clarified.

Reviewer 2 Report

The manuscript reports valuable experimental results, which can be interesting for potential readers of the Journal. This submission is recommended for publication in Materials after minor modification. The following comments and suggestions would help improving quality of this work:

1) Title is too long. The term “high performance” is uninformative without a clarification. The acronym “SCC” is commonly accepted. Thus, the reviewer suggest the following title alternative: “Rheological behavior and microstructure characteristics of SCC incorporating metakaolin and silica fume”.

2) Abstract:

The statement “the results show that addition of MK and SF show negative effect on the flowability in general, with exception (?) of replacement ratio of 2% (?)” is unclear. The      wording “show (shows) negative effect on the flowability” is imperfect. (The reviewer recommends the formulation “addition of MK and SF reduces flowability” as an appropriate alternative.) The term “replacement ratio” must be described. What was the material replaced? Most importantly, the statement “with exception” requires a clarification. Why the indicated replacement ratio has the exceptional effect on the flowability? The same comment is related to Section 3.2 and Conclusions.

The sentences “A higher yield stress is found to negatively affect fluidity, slump flow and SA based on rheology and workability results” and “…compounding appropriate (?) SF and MK” are unclear and stylistically imperfect.

3) Introduction must describe novelty of the research in context of existing knowledge in the field. What is the novelty aspect of this study? Physical nature of the combined application efficiency of metakaolin and silica fume must be clarified with the reference to information reported in the literature. Literature review must also substantiate the choice of the proportions of the concrete mixtures used in this study. The statement “optimized (?) mechanical properties” (Line 76) requires a clarification. The term “optimized” is evidently unsuitable in the context of this study: none of optimization procedures were applied.

4) Section 2:

Line 83. A sentence should not begin with a number.

Table 2. Please, check the units (“g”). The term “mix design” is misleading. The latter comment is also related to Table 3.

Lines 113-114. The explanation “m1 and m2 replace (?) the weight of dried aggregate in bottom and top (?)” is unclear.

Line 133. SI units must be used.

5) Section 3:

Lines 137-138. The wording “It is because that…” is imperfect stylistically. The statement “specific surface area of MK and SF is obviously (where is it shown?) larger than that of fly ash” must be related to a figure and/or referred to a literature source.

Lines 160-162. The sentence “When the content of MK and SF is 2%, the viscosity and yield stress decrease obviously (where is it shown?), this is due to the small particle size of MK and SF” must be clarified. The reader must find and interpret the data himself. That is unacceptable. What are the results compared here? The percentage of MK and SF must be indicated in Table 4. It should be also pointed out that the particle size might explain an      alteration of the material characteristics only indirectly. A more detail discussion of this phenomenon is necessary. Results presented in Sections 3.4 and 3.5 could be briefly introduced here.

Lines 171-172. The sentence “This is because when the two (?) are used in combination, the accumulation of the powder becomes closer (?)” is unclear and stylistically imperfect.

Lines 183-196. This text must be rewritten. The analysis must be related with the corresponding test results of particular specimens. For example, the statement “Both MK and SF will increase the fluidity and segregation of SCC when their content is 2%” is unclear. What are the considered specimens? The same comment is related to the statement “With the incorporation of SF and MK over 6%, the working performance of SCC will not meet the requirements of self-compacting concrete”. Where is it shown? The term “working performance” is unusual. The reviewer also recommends indicating the workability limit in Figure 6. The explanation “…this is attributed to the further (?) densification of the accumulation (?) of particles…” is stylistically imperfect and unclear. The statement “…result in a better (?) water reduction effect” is unclear. Where are the corresponding test results shown? The term “over-high fluidity” is uncommon. The statement “the mixture of SF and MK should be controlled within 6% without increasing the SP dosage” is far too categorical. It can be considered only as a recommendation that was defined by analyzing a limited number of test specimens.

Lines 200-203. This text requires modifications. The wording “correlation between them (?)” is unclear: the correlated variables must be defined. The statement “This (?) shows” is      also unclear. The sentence “The reason is that when the driving force governing flow is greater than the yield stress of SCC, the flow continues; and when these two parameters reaches an equilibrium, the flow stops” must be reformulate as stylistically imperfect. A literature reference must also support this statement.

Lines 206-214. This discussion must be noticeably extended; the writing style should be      improved as well. The statements “linear relationship can also be seen between them” and “it is not completely linear relationship” are contradicting each other. The sentence “This (?) indicate (indicates) that SA is also affected by viscosity of concrete, this (?) is mainly caused by vibration condition (?) during SA testing process” is unclear. What is the physical nature of the observed relationship? The SA testing procedure must be described in more detail. What does the “vibration condition”      describe? The text “Although yield stress is the determinant of aggregate stability (?), but it (?) will be significantly reduced (Where is it shown?) under vibration conditions (?), as a result, the aggregate fails to remain suspended and sinks (Where is it shown?). At this point (?)” is unclear. The statement “the performance of SCC can be predicted to a certain extent by evaluating the rheology property of paste” is not supported by the reported results. A reference to literature might be useful here.

Lines 218-234. This text requires improvements. The wording “…the addition of MK and SF can significantly increase the compressive strength of SCC. For example…” requires a clarification. What are the specimens compared here? Moreover, the limited number of the test samples (three cubes) does not enable estimating variation (i.e. significance) of the test outcomes. That makes application of the term “significantly” inappropriate in this context. The statement “…compressive strengths are further (?) increased” is unclear. The explanation “It is because that the particle size of MK and SF are smaller than FA” is also imperfect. It must be referred to Figure 1 and, possibly, to the outcomes presented in Sections 3.4 and 3.5, though only indirect relationship between the particle size and compressive strength can be identified using the reported test results. A literature reference can be also useful here. The adjective “higher” (Line 226) is inappropriate, since it requires a reference for the comparison. The text “The enhancement effect will further (?) increase when MK and SF are binary applied. As the addition of MK and SF is 8% (What are the specimens considered?), the strength of SCC decreases, which is due to the low fluidity of SCC and the difficulty of releasing bubbles (What does it mean?), this further (?) illustrates the importance of workability to the mechanical properties of SCC. In order to further (?) expound…” must be rewritten.

Line 240. The statement “The employment of SF decreases the formation of AFt” must be clarified. What are the specimens (results) considered here?

Line 242. The note “It can be clearly observed…” is unclear. What are the results referred with?

Lines 244-246. The sentence “…more Ca(OH)2 produced by cement hydration is consumed by pozzolanic reaction and better performances such as mechanical properties can be resulted” requires a clarification. What are the test results considered here?

Line 255. The note “It can be clearly found…” is also unclear. What are the results referred with? If it was shown in literature, this sentence must be reformulated appropriately. If not, the corresponding test results/specimens must be defined properly.

Lines 259, 260. The statements “…accordant with the XRD results” and “…samples doped with MK” must be referred with particular test outcomes.

Lines 269-270. A literature reference must support the statement “the employment of MK and SF optimizes the pores structures of SCC by reducing the total porosity and the volume of coarse pores.

Lines 271, 274, 275. The wordings “In addition, 6% addition”, “the MK and SF are smaller (?)”, and “the gaps of mixtures (?)” are stylistically imperfect.

Line 281. The wording “samples contain 6% addition MK” is stylistically imperfect and unclear. What are the specimens considered here?

6) Conclusions. The statement “The employment of MK and SF reduce (reduces) the fluidity of SCC with the exception (?) of replacement ratio of 2%” (Lines 295-296) requires an explanation. What was the material replaced? What is the physical origin of the exceptional outcome? A similar comment is related to the sentence “The beneficial effect is more significant when they are used together.” A physical nature of the observed efficiency of the combined application of MK and SF must be described.

7) References. The formatting style of the list must be unified: the references [1], [3], [14], [16], and [30]-[33] give examples of different format.

Author Response

Response to Reviewer 2 Comments

Piont 1: Title is too long. The term “high performance” is uninformative without a clarification. The acronym “SCC” is commonly accepted. Thus, the reviewer suggest the following title alternative: “Rheological behavior and microstructure characteristics of SCC incorporating metakaolin and silica fume”.

Response 1: Thank you for your valuable comment. The review is right. In the revised manuscript, the title has been checked according to the reviewer’s advice.

Piont 2:  Abstract:

The statement “the results show that addition of MK and SF show negative effect on the flowability in general, with exception (?) of replacement ratio of 2% (?)” is unclear. The wording “show (shows) negative effect on the flowability” is imperfect. (The reviewer recommends the formulation “addition of MK and SF reduces flowability” as an appropriate alternative.) The term “replacement ratio” must be described. What was the material replaced? Most importantly, the statement “with exception” requires a clarification. Why the indicated replacement ratio has the exceptional effect on the flowability? The same comment is related to Section 3.2 and Conclusions.

The sentences “A higher yield stress is found to negatively affect fluidity, slump flow and SA based on rheology and workability results” and “…compounding appropriate (?) SF and MK” are unclear and stylistically imperfect.

Response 2: Thank you for your useful comment. The review is right. In the revised manuscript, the statement “the results show that addition of MK and SF show negative effect on the flowability in general, with exception of replacement ratio of 2%” has been illuminated, the appropriate formulation “addition of MK and SF reduces flowability” has been cited. Then, the statement “with exception” has been clearly explained, as well as the statement related to Section 3.2 and Conclusions.

The sentences “A higher yield stress is found to negatively affect fluidity, slump flow and SA based on rheology and workability results” and “…compounding appropriate SF and MK” have been checked and clarified.

Piont 3: Introduction must describe novelty of the research in context of existing knowledge in the field. What is the novelty aspect of this study? Physical nature of the combined application efficiency of metakaolin and silica fume must be clarified with the reference to information reported in the literature. Literature review must also substantiate the choice of the proportions of the concrete mixtures used in this study. The statement “optimized (?) mechanical properties” (Line 76) requires a clarification. The term “optimized” is evidently unsuitable in the context of this study: none of optimization procedures were applied.

Response 3: Thank you for your useful comment. The review is right. In the revised manuscript, novelty of the research in context of existing knowledge in the field has been described in line 74-79. The literature has been appended to clarify physical nature of the combined application efficiency of metakaolin and silica fume.

[Song Q, Yu R, Wang X, et al. A novel self-compacting ultra-high performance fibre reinforced concrete (SCUHPFRC) derived from compounded high-active powders. Constr. Build. Mater., 2018, 158: 883-893.]

In the conclusions of this literature, it is pointed out that the effect of hybrid use of SF and MK on UHPFRC properties can be attributed to physical and chemical influences. On the one hand, the utilized MK is more efficient than SF in promoting the hydration kinetics of UHPFRC cementitious system. On the other hand, excess amount of MK can enhance the shrinkage and viscosity of UHPFRC mixture, which result in that its microstructure development may be disturbed by shrinkage caused micro-cracks and trapped bubbles. Hence, to develop a SCUHPFRC with advanced properties, the use of appropriate compounded high-active powders is a crucial factor.

As a result, it is logical and novel to study the feasibility of producing SCC by composite use of MK and SF.

Then a literature review which substantiates the choice of the proportions of the concrete mixtures used in this study has been supplemented and clarified, as shown in line 62-63, it is found that SCCs contain 30-60% FA have good mechanical properties and durability in this literature, and the dosage in this study is 45%. In addition, the contents of MK and SF is about 8% in ultra-high-performance concrete (UHPC), as shown in line 76-77. The max dosage of MK and SF is 8% in this study.

[Yazici H. The effect of silica fume and high-volume Class C fly ash on mechanical properties, chloride penetration and freeze-thaw resistance of self-compacting concrete. Constr. Build. Mater., 2008, 22(4):456-462.]

In addition, the term “optimized” has been checked.

Piont 4: Section 2:

Line 83. A sentence should not begin with a number.

Table 2. Please, check the units (“g”). The term “mix design” is misleading. The latter comment is also related to Table 3.

Lines 113-114. The explanation “m1 and m2 replace (?) the weight of dried aggregate in bottom and top (?)” is unclear.

Line 133. SI units must be used.

Response 4: Thank you for your useful comment. The review is right.

Line 83. The sentence has been rewritten in the revised manuscript.

Table 2. The units (“g”) have been checked in the revised manuscript.

Lines 113-114. The explanation “m1 and m2 replace the weight of dried aggregate in bottom and top” has been checked and replaced by “m1 and m2 are refer to the weight of dried aggregates in bottom and top layer of segregation barrel” in the revised manuscript.

Line 133. The units (“psi”) have been replaced by SI units (“MPa”) in the revised manuscript.

Piont 5: Section 3:

Lines 137-138. The wording “It is because that…” is imperfect stylistically. The statement “specific surface area of MK and SF is obviously (where is it shown?) larger than that of fly ash” must be related to a figure and/or referred to a literature source.

Lines 160-162. The sentence “When the content of MK and SF is 2%, the viscosity and yield stress decrease obviously (where is it shown?), this is due to the small particle size of MK and SF” must be clarified. The reader must find and interpret the data himself. That is unacceptable. What are the results compared here? The percentage of MK and SF must be indicated in Table 4. It should be also pointed out that the particle size might explain an alteration of the material characteristics only indirectly. A more detail discussion of this phenomenon is necessary. Results presented in Sections 3.4 and 3.5 could be briefly introduced here.

Lines 171-172. The sentence “This is because when the two (?) are used in combination, the accumulation of the powder becomes closer (?)…” is unclear and stylistically imperfect.

Lines 183-196. This text must be rewritten. The analysis must be related with the corresponding test results of particular specimens. For example, the statement “Both MK and SF will increase the fluidity and segregation of SCC when their content is 2%” is unclear. What are the considered specimens? The same comment is related to the statement “With the incorporation of SF and MK over 6%, the working performance of SCC will not meet the requirements of self-compacting concrete”. Where is it shown? The term “working performance” is unusual. The reviewer also recommends indicating the workability limit in Figure 6. The explanation “…this is attributed to the further (?) densification of the accumulation (?) of particles…” is stylistically imperfect and unclear. The statement “…result in a better (?) water reduction effect” is unclear. Where are the corresponding test results shown? The term “over-high fluidity” is uncommon. The statement “the mixture of SF and MK should be controlled within 6% without increasing the SP dosage” is far too categorical. It can be considered only as a recommendation that was defined by analyzing a limited number of test specimens.

Lines 200-203. This text requires modifications. The wording “correlation between them (?)” is unclear: the correlated variables must be defined. The statement “This (?) shows” is also unclear. The sentence “The reason is that when the driving force governing flow is greater than the yield stress of SCC, the flow continues; and when these two parameters reaches an equilibrium, the flow stops” must be reformulate as stylistically imperfect. A literature reference must also support this statement.

Lines 206-214. This discussion must be noticeably extended; the writing style should be improved as well. The statements “linear relationship can also be seen between them” and “it is not completely linear relationship” are contradicting each other. The sentence “This (?) indicate (indicates) that SA is also affected by viscosity of concrete, this (?) is mainly caused by vibration condition (?) during SA testing process” is unclear. What is the physical nature of the observed relationship? The SA testing procedure must be described in more detail. What does the “vibration condition” describe? The text “Although yield stress is the determinant of aggregate stability (?), but it (?) will be significantly reduced (Where is it shown?) under vibration conditions (?), as a result, the aggregate fails to remain suspended and sinks (Where is it shown?). At this point (?)…” is unclear. The statement “the performance of SCC can be predicted to a certain extent by evaluating the rheology property of paste” is not supported by the reported results. A reference to literature might be useful here.

Lines 218-234. This text requires improvements. The wording “…the addition of MK and SF can significantly increase the compressive strength of SCC. For example…” requires a clarification. What are the specimens compared here? Moreover, the limited number of the test samples (three cubes) does not enable estimating variation (i.e. significance) of the test outcomes. That makes application of the term “significantly” inappropriate in this context. The statement “…compressive strengths are further (?) increased” is unclear. The explanation “It is because that the particle size of MK and SF are smaller than FA” is also imperfect. It must be referred to Figure 1 and, possibly, to the outcomes presented in Sections 3.4 and 3.5, though only indirect relationship between the particle size and compressive strength can be identified using the reported test results. A literature reference can be also useful here. The adjective “higher” (Line 226) is inappropriate, since it requires a reference for the comparison. The text “The enhancement effect will further (?) increase when MK and SF are binary applied. As the addition of MK and SF is 8% (What are the specimens considered?), the strength of SCC decreases, which is due to the low fluidity of SCC and the difficulty of releasing bubbles (What does it mean?), this further (?) illustrates the importance of workability to the mechanical properties of SCC. In order to further (?) expound…” must be rewritten.

Line 240. The statement “The employment of SF decreases the formation of AFt” must be clarified. What are the specimens (results) considered here?

Line 242. The note “It can be clearly observed…” is unclear. What are the results referred with?

Lines 244-246. The sentence “…more Ca(OH)2 produced by cement hydration is consumed by pozzolanic reaction and better performances such as mechanical properties can be resulted” requires a clarification. What are the test results considered here?

Line 255. The note “It can be clearly found…” is also unclear. What are the results referred with? If it was shown in literature, this sentence must be reformulated appropriately. If not, the corresponding test results/specimens must be defined properly.

Lines 259, 260. The statements “…accordant with the XRD results” and “…samples doped with MK” must be referred with particular test outcomes.

Lines 269-270. A literature reference must support the statement “the employment of MK and SF optimizes the pores structures of SCC by reducing the total porosity and the volume of coarse pores.”

Lines 271, 274, 275. The wordings “In addition, 6% addition…”, “the MK and SF are smaller (?)”, and “the gaps of mixtures (?)” are stylistically imperfect.

Line 281. The wording “samples contain 6% addition MK” is stylistically imperfect and unclear. What are the specimens considered here?

Response 5: Thank you for your elaborate comments and positive suggestions. The review is right.

Lines 137-138. In the revised manuscript, the wording “It is because that…” has been rewritten. The specific surface area of MK, SF and FA have been supplemented and shown in Table 1.

Lines 160-162. In the revised manuscript, the sentence “When the content of MK and SF is 2%, the viscosity and yield stress decrease obviously, this is due to the small particle size of MK and SF” has been and written, as shown in 175-179. The analysis of raw data in this section has been supplemented to clarify the result. The percentage of MK and SF has been indicated in Table 4. Moreover, results presented in Sections 3.4 and 3.5 have been introduced to expound more detail discussion of this phenomenon, as shown in 187-187.

Lines 171-172. In the revised manuscript, the sentence “This is because when the two are used in combination, the accumulation of the powder becomes closer …” has been checked and rewritten, as shown in 192-194.

Lines 183-196. In the revised manuscript, this text has been checked carefully and rewritten according the reviewer comments, as shown in line 204-225. The corresponding test results of particular specimens for each analysis have been spelled out, as shown in 205-208, 212-214 and 221-224. The term “working performance” has been replaced by “workability”. The workability limit of SCC has been added in line 224 and Figure 6. The format and grammar of explanation “…this is attributed to the further densification of the accumulation of particles…” has been modified, and the statement “…result in a better water reduction effect” has been illuminated as shown in line 215-218. The term “working performance” has been rectified. The statement “the mixture of SF and MK should be controlled within 6% without increasing the SP dosage” has been refined and written in a more appropriate way, as shown in line 224-225.

Lines 200-203. In the revised manuscript, the text has been modified. The wording “correlation between them …” has been rewritten, the correlated variables have been defined as “it shows a significant linear relationship between yield stress and slump flow, the slump flow decreases with the increase of yield stress”, as shown in line 228-230. The statement “This shows” has been replaced “This phenomenon illustrates”. Then the sentence “The reason is that when the driving force governing flow is greater than the yield stress of SCC, the flow continues; and when these two parameters reaches an equilibrium, the flow stops” has been checked, as shown in 231-232. In addition, a literature reference has been added to support this statement.

[Wu Q, An X. Development of a mix design method for SCC based on the rheological characteristics of paste. Constr. Build. Mater., 2014, 53(3): 642-651.]

The difference between the gravitational and buoyant forces must exceed the sum of the braced forces generated by the cells below to ensure its movement.

Lines 206-214. In the revised manuscript, this discussion has been checked carefully and extended in detail, as shown in line 235-246. In particular, the physical nature of the observed relationship has been expounded, and the SA testing procedure and “vibration condition” has been described in more detail, as shown in line 121-125, . In addition, 3 reference literatures have been cited here to support the statements.

[Schowalter W R, Christensen G. Toward a rationalization of the slump test for fresh concrete: Comparisons of calculations and experiments. Journal of Rheology, 1998, 42(4):865-870.].

[Wan B, Gadalamaria F, Petrou M F. Influence of mortar rheology on aggregate settlement. ACI Mater. J., 2000, 97(4): 479-485.].

Fresh concrete exhibits a yield stress that, under certain conditions, prevents the settlement of coarse aggregate, although its density is larger than that of the suspending mortar. The stone does not sink in the undisturbed mortar (which has a high yield stress), but sinks when the mortar is vibrated, presumably due to a large reduction in its yield stress.

[Ng I Y T, Ng P L, Kwan A K H. Rheology of Mortar and its Influences on Performance of Self-Consolidating Concrete. Key Engineering Materials, 2008, 400-402:421-426.].

Using a SCM mix with saturation SP dosage as the mortar phase can produce SCC with high performance and therefore is a good starting point to optimize the performance of SCC mixes.

Lines 218-234. In the revised manuscript, this text has been improved carefully according to the comments of reviewer. First of all, the compared specimens for each discussion is clarified, as shown in line 255-258 and 270-271. Secondly, the inappropriate words “significantly” and “higher” have been replaced, the improper sentences “compressive strengths are further (?) increased” and “The enhancement effect will further (?) increase have been rewritten. The explanation “It is because that the particle size of MK and SF are smaller than FA” has been clarified, and the outcomes presented in Sections 3.4 and 3.5 has been adopted here, as shown in 262-268. In the end, a literature reference has been added.

[Song Q, Yu R, Wang X, et al. A novel self-compacting ultra-high performance fibre reinforced concrete (SCUHPFRC) derived from compounded high-active powders. Constr. Build. Mater., 2018, 158: 883-893.]

Line 240. In the revised manuscript, the statement “The employment of SF decreases the formation of AFt” has been clarified and shown as “The diffraction peak of AFt in PSF6 is lower than that of P0, which indicates that the employment of SF decreases the formation of AFt”, as shown in line 281-282.

Line 242. In the revised manuscript, the referred results has been supplemented for the noteIt can be clearly observed…”, as shown in line 284-286.

Lines 244-246. In the revised manuscript, the reference results for the sentence “…more Ca(OH)2 produced by cement hydration is consumed by pozzolanic reaction and better performances such as mechanical properties can be resulted” have been clarified, as shown in line 286-290, this statement is based on the results of compressive strength test.

Line 255. In the revised manuscript, the corresponding test results and specimens have been defined properly, the sentence has been rewritten, as shown in line 299-231.

Lines 259, 260. In the revised manuscript, the referred tests outcomes for the statements “…accordant with the XRD results” and “…samples doped with MK” have been supplemented, as shown in 302-305.

Lines 269-270. In the revised manuscript, a literature reference has been appended to support the statement “the employment of MK and SF optimizes the pores structures of SCC by reducing the total porosity and the volume of coarse pores.”, as shown in line 316.

[Duan P, Shui Z H, Chen W, et al. Effects of metakaolin, silica fume and slag on pore structure, interfacial transition zone and compressive strength of concrete. Constr. Build. Mater., 2013, 44(7): 1-6.]

Lines 271, 274, 275. In the revised manuscript, the wordings “In addition, 6% addition…”, “the MK and SF are smaller (?)”, and “the gaps of mixtures (?)” have been improved, as shown in line 317, 320 and 321-322.

Line 281. In the revised manuscript, the wording “samples contain 6% addition MK” has been improved. The specimens considered is CMK6 and has been indicated as shown in 326.

Piont 6: Conclusions. The statement “The employment of MK and SF reduce (reduces) the fluidity of SCC with the exception (?) of replacement ratio of 2%” (Lines 295-296) requires an explanation. What was the material replaced? What is the physical origin of the exceptional outcome? A similar comment is related to the sentence “The beneficial effect is more significant when they are used together.” A physical nature of the observed efficiency of the combined application of MK and SF must be described.

Response 6: Thank you for your useful comments. The review is right. In the revised manuscript, the statement “The employment of MK and SF reduce (reduces) the fluidity of SCC with the exception (?) of replacement ratio of 2%” (Lines 295-296) and “The beneficial effect is more significant when they are used together.” have been rewritten and explained in detail. The physical nature of the observed efficiency of the combined application of MK and SF has been illuminated, as shown in 357-363.

Piont 7: References. The formatting style of the list must be unified: the references [1], [3], [14], [16], and [30]-[33] give examples of different format.

Response 7: Thank you for your useful comments. The review is right. In the revised manuscript, the formatting style of references has been unified.

Reviewer 3 Report

The paper concerns the investigations about the rheological properties of self-compacting concrete with the addition of metakaolin and silica fume. The studies are supplemented with microstructure analysis, which helps to interpret the rheological data obtained. The study presented has a high scientific value and presents actual topic in terms of main research directions in the concrete technology. Experiment is well designed and executed. After all, the paper requires major revision before acceptance for publication. Detailed comments are listed below:

1.      There are few language, syntax errors. I suggest checking the article by English native speaker; e.g. line 225 – it is "powers", it guess it should be "powders".

2.      In addition to the particle size distribution curve also an important information is the specific surface area of the binders used, especially when the rheological parameters are tested and interpreted. Please provide such information if it is known.

3.      Table 1 – please indicate by what method the chemical composition has been determined. If it is data already published, please cite a reference to a specific item in the literature.

4.      Table 2 - it would be more appropriate to provide data in kg per 1m3 of paste, not in [g] where it is not exactly stated to what volume of paste it refers to.

5.      There is a few linear equations in the paper provided – line 152, 176, 199, 207 – please specify the method by which the given equations were calculated.

6.      Line 176 – there is placed the reference to the Fig. 5 and the equation. This equation is different than that which is shown in this figure. Please correct this, the same is with the R2 coefficient.

7.      Line 242-246 – authors presents the conclusions about the hydration process of cement with different supplements. It is well known in the literature and it can be confirmed by the research of other authors. Examples of papers that can be referred to and which confirm your reports:

Szeląg M.: Development of cracking patterns in modified cement matrix with microsilica. Materials, vol. 11(10), 2018, s. 1928

Poon, C. S., Kou, S. C., & Lam, L. (2006). Compressive strength, chloride diffusivity and pore structure of high performance metakaolin and silica fume concrete. Construction and building materials, 20(10), 858-865.

Nili, M., & Ehsani, A. (2015). Investigating the effect of the cement paste and transition zone on strength development of concrete containing nanosilica and silica fume. Materials & Design, 75, 174-183.

Author Response

Response to Reviewer 3 Comments

Piont 1: There are few language, syntax errors. I suggest checking the article by English native speaker; e.g. line 225 – it is "powers", it guess it should be "powders".

Response 1: Thank you for your useful comment. The review is right. In the revised manuscript, the article has been checked carefully, the language and syntax errors have been rectified.

Piont 2: In addition to the particle size distribution curve also an important information is the specific surface area of the binders used, especially when the rheological parameters are tested and interpreted. Please provide such information if it is known.

Response 2: Thank you for your valued comment. The review is right. In the revised manuscript, the information of specific surface area for cement, FA, MK and SF has been provided and shown in Table 1.

Piont 3: Table 1 – please indicate by what method the chemical composition has been determined. If it is data already published, please cite a reference to a specific item in the literature.

Response 3: Thank you for your useful comment. The review is right. The chemical composition in Table 1 has been determined by X-ray fluorescence spectrometer, and it has been indicated in the revised manuscript.

Piont 4: Table 2 - it would be more appropriate to provide data in kg per 1m3 of paste, not in [g] where it is not exactly stated to what volume of paste it refers to.

Response 4: Thank you for your valuable comment. The review is right. In the revised manuscript, the unit of “g” in Table 2 has been checked, and “kg/m3” has been adopted.

Piont 5: There is a few linear equations in the paper provided – line 152, 176, 199, 207 – please specify the method by which the given equations were calculated.

Response 5: Thank you for your valuable comment. The review is right. The Origin Pro software is employed to perform the procedure the linear fitting, the formulas in line 152, 176, 199, 207 are based on this method. It has been clarified in the revised manuscript.

Piont 6: Line 176 – there is placed the reference to the Fig. 5 and the equation. This equation is different than that which is shown in this figure. Please correct this, the same is with the R2 coefficient.

Response 6: Thank you for your valuable comment. The review is right. In the revised manuscript, the equation and R2 coefficient has been corrected and in accordance with Fig. 5.

Piont 7: Line 242-246 – authors presents the conclusions about the hydration process of cement with different supplements. It is well known in the literature and it can be confirmed by the research of other authors. Examples of papers that can be referred to and which confirm your reports:

Szeląg M.: Development of cracking patterns in modified cement matrix with microsilica. Materials, vol. 11(10), 2018, s. 1928

Poon, C. S., Kou, S. C., & Lam, L. (2006). Compressive strength, chloride diffusivity and pore structure of high performance metakaolin and silica fume concrete. Construction and building materials, 20(10), 858-865.

Nili, M., & Ehsani, A. (2015). Investigating the effect of the cement paste and transition zone on strength development of concrete containing nanosilica and silica fume. Materials & Design, Mater. Design 75, 174-183.

Response 7: Thank you for your useful comment. The review is right. These literatures have been read in-depth and added in the revised manuscript, to confirmed the conclusions about the hydration process of cement with different supplements in line 242-246.

Round 2

Reviewer 3 Report

All suggestions have been included. I accept the paper for publication.